# Manganese Porphyrin Reduces Oxidative Stress in Vulnerable Parkin-Null *Drosophila* Dopaminergic Neurons

**DOI:** 10.3390/antiox14091031

**Published:** 2025-08-22

**Authors:** Amber N. Juba, Petros P. Keoseyan, Riley P. Hamel, Tigran Margaryan, Michaela L. Barber, Amanda N. Foley, T. Bucky Jones, Ines Batinic-Haberle, Artak Tovmasyan, Lori M. Buhlman

**Affiliations:** 1Biomedical Sciences Program, Midwestern University, Glendale, AZ 85308, USA; ajuba@midwestern.edu (A.N.J.); michaela.barber@midwestern.edu (M.L.B.); 2Arizona College of Osteopathic Medicine, Midwestern University, Glendale, AZ 85308, USA; pkeoseyan78@midwestern.edu (P.P.K.); riley.hamel@midwestern.edu (R.P.H.); amanda.foley@midwestern.edu (A.N.F.); bjones1@midwestern.edu (T.B.J.); 3Department of Translational Neuroscience, Ivy Brain Tumor Center, Barrow Neurological Institute, Phoenix, AZ 85013, USA; tigran.margaryan@ivybraintumorcenter.org (T.M.); artak.tovmasyan@barrowneuro.org (A.T.); 4Department of Radiation Oncology, Duke University Medical Center, Durham, NC 27710, USA; ibatinic@duke.edu

**Keywords:** manganese porphyrin, parkin, oxidative stress, antioxidants, mitochondria, hydrogen peroxide, glutathione, dopaminergic neuron, *Drosophila*, redox

## Abstract

Oxidative stress and mitochondrial dysfunction are heavily implicated in all forms of Parkinson’s disease; however, antioxidant administration has largely failed in clinical trials. Among the likely causes of failure are brain bioavailability and cellular redox state. We have administered two manganese porphyrin compounds with different bioavailability, MnTE-2-PyP^5+^ and MnTnBuOE-2-PyP^5+^, to parkin-null *Drosophila* food and found that the more bioavailable one, with higher brain and mitochondrial availability, MnTnBuOE-2-PyP^5+^, improves developmental deficits and motivated behavior in female flies. Using highly sensitive redox reporters, we further found that MnTnBuOE-2-PyP^5+^ reduces hydrogen peroxide levels in mitochondria of dopaminergic neurons that are functionally homologous to the mammalian substantia nigra and facilitates motivated behavior in female flies. Interestingly, both compounds reduce an oxidative stress marker at the whole-brain level and extend lifespan in control flies. Neither compound improves lifespan in parkin-null flies. Thus, additional studies, changing the timing and/or dosage of compound administration, are warranted.

## 1. Introduction

Parkin is an E3 ubiquitin ligase that targets mitochondrial proteins for proteasomal or lysosomal digestion [1,2,3,4,5]. Loss of parkin function causes a rare form of Parkinson’s disease (PD), and in *Drosophila*, it causes developmental deficiencies, decreased lifespan, deficits in motivated behavior, and selective degeneration of protocerebral posterior lateral region 1 (PPL1) dopaminergic neurons that are functionally homologous to the mammalian substantia nigra [6,7,8,9,10,11]. Thus, *Drosophilae* are a powerful tool regarding the search for mechanisms of dopaminergic degeneration in the absence of parkin. Further, oxidative stress and mitochondrial dysfunction are primarily implicated in the neurodegeneration that occurs in the absence of parkin, as well as in the more common idiopathic forms of PD [12,13,14,15,16,17,18,19]. We previously reported that protein oxidation, hydrogen peroxide levels, and glutaredoxin (Grx) activity are elevated in PPL1 mitochondria of parkin-null flies [8,20]. Strategies to reduce reactive oxygen species (ROS) levels or increase antioxidant activity have therapeutic potential for all forms of PD; however, antioxidant therapies fail to show promise in clinical trials [21,22,23,24,25,26,27,28,29]. Among several plausible reasons is the inability of pharmaceuticals to access affected neurons and/or their subcellular targets like mitochondria [30,31,32,33]. Additionally, previously tested compounds may have inefficient redox activity and/or they may fail to target relevant ROS or redox reactions [34].

Manganese porphyrin compounds (MnPs) are redox-active compounds that were designed as potent mimics of superoxide dismutase (SOD) [35,36,37,38,39]. Selected MnPs, MnTE-2-PyP^5+^ (MnE) and MnTnBuOE-2-PyP^5+^ (MnBuOE) are the current most optimized molecules that have shown significant potency in multiple preclinical disease models, including neurodegenerative disorders, cancer, and ischemia–reperfusion injury [35,36]. Mechanistically, both MnPs catalyze the dismutation of superoxide into hydrogen peroxide and oxygen with high efficiency; hydrogen peroxide is further metabolized by catalase or peroxidases to water and oxygen. Both MnPs are also capable of scavenging other reactive species including peroxynitrite, hydrogen peroxide, hypochlorite and carbonate anion radicals [40]. Additionally, MnPs utilize hydrogen peroxide/glutathione (G-SH) to facilitate protein glutathionylation and modulate cell survival and stress adaptation signaling pathways, such as nuclear factor–kappa B (NF-kB) and NF-E2-related factor 2 (Nrf2)/Kelch-like-ECH-associated protein 1 (Nrf2/Keap1) [40]. Importantly, MnE and MnBuOE were shown to accumulate in mouse heart mitochondria at 1.6-fold and 3-fold higher ratios relative to the cytosol, respectively [35]. However, only MnBuOE was able to enter brain mitochondria at 2:1, mitochondria to cytosol ratio, while MnE was found in brain cytosol but not in brain mitochondria [35,41]. The brain and, more so, the brain’s mitochondrial accumulation are driven by their relative lipophilicities, the presence of pentacationic charges on the molecule, and their 3-dimentional structure, with MnBuOE demonstrating enhanced brain and brain mitochondrial penetration as compared to MnE [42]. These optimized properties and potent redox-abilities support the significant therapeutic potential of these MnPs in diseases characterized by oxidative stress, and are therefore currently being tested in several clinical trials [40]. The purpose of the present study was to examine the ability of MnE and MnBuOE to reduce oxidative stress at the whole-brain level and in PPL1 mitochondria of control and parkin-null *Drosophila* [6,35].

## 2. Materials and Methods

### 2.1. Chemicals and Reagents

MnTE-2-PyP^5+^ (MnE, manganese(III) *meso*-tetrakis(*N*-ethylpyridinium-2-yl)porphyrin) and MnTnBuOE-2-PyP^5+^ (MnBuOE, manganese(III) *meso*-tetrakis(*N*-butoxyethylpyridinium-2-yl)porphyrin) were synthesized, purified, and characterized as previously described [43]. Liquid chromatography-mass spectrometry (LC-MS)-grade water was obtained from a Milli-Q IQ 7000 filtration system (Millipore Sigma, Burlington, MA, USA). L-cysteinylglycine, DL-cystine-d_6_, glutathione (glycine-^13^C_2_,^15^N) sodium salt, D,L-cystathionine-d_4_, and L-cysteine-^13^C_3_,^15^N, were obtained from Toronto Research Chemicals (Toronto, ON, Canada). L-glutathione (GSH), L-cysteine hydrochloride, L-methionine, L-cystathionine, L-cystine, and *N*-ethylmaleimide (NEM), 5-Sulfosalicylic Acid (SSA), were obtained from Millipore Sigma (Burlington, MA, USA). L-γ-glutamyl-L-cysteine (ammonium salt) and L-methionine-d_3_ were purchased from Cayman Chemical (Ann Arbor, MI, USA). LC-MS-grade methanol, formic acid (>98% grade), HPLC-grade ammonium formate, and acetonitrile were purchased from Fisher Scientific (Waltham, MA, USA).

### 2.2. Drosophila Maintenance and MnP Administration

*Drosophila* were maintained at 25 °C in 12 h of light and constant humidity and transferred to fresh food vials every three or four days. Parent stocks were placed on standard cornmeal and molasses food supplemented with 0 or 10 µM MnE or MnBuOE. Fertilized embryos were deposited on the food so that progeny larvae consumed MnP throughout development. Adult progeny were collected within 48 h of eclosion and maintained on fresh supplemented food vials; flies were transferred to fresh supplemented food vials every three to four days. Control flies (*w^1118^*) were obtained from Bloomington Stock Center (Bloomington, IN, USA) and parkin loss-of-function mutant stocks (harboring the *park^25^* mutation) were a gift from Leo Pallanck at the University of Washington, Seattle (described in [6]). Parkin-mutant flies were backcrossed with control for developmental, thiol/disulfide analyses, climbing, and survival studies. Control genotype is *w^1118^*; *+^w1118^*; *+^w1118^*, where “*+^w1118^*” indicates a chromosome from the control *w^1118^* stock. Parkin-null genotype is *w^1118^*; +*^w1118^*; *park^25^*/*park^25^*. For PPL1 hydrogen peroxide and GSH redox equilibrium experiments, control and parkin-null flies expressed redox-sensitive green fluorescent protein 2 (GFP2) fused to *C. elegans* oxidant receptor peroxidase 1 (Orp1) or human Grx1 (mito-roGFP2-Orp1 or mito-roGFP2-Grx1, respectively) under the control of tyrosine hydroxylase (TH) expression using the GAL4>UAS expression system (*THGAL4>UAS-mito-roGFP2-Grx1*/*Orp1*) [44,45,46,47,48]. A mitochondrial localizing sequence efficiently directs the reporters to the mitochondrial matrix. *TH-GAL4* and *RoGFP2* stocks were purchased from the Bloomington Stock Center (BL# 67664, 67667, and 8848). The control hydrogen peroxide reporter genotype is *w^1118^*; *+^w1118^*/*UAS-mito-roGFP2-Orp1*; *+^w1118^*/*TH-GAL4*. Parkin-null genotype is *w^1118^*; *+^w1118^*/*UAS-mito-roGFP2-Orp1*; *TH-GAL4*, *park^25^*/*park^25^*. GSH redox equilibrium reporting genotypes are similar, except *UAS-mito-roGFP2* is fused to *Grx1*.

### 2.3. Development Assay

Twenty third instar larva were placed in food vials supplemented with 0, 2, or 10 µM MnBuOE, and vials were monitored daily for pupariation (pupa case formation) and eclosion (emergence of adult flies from pupa cases). The time required for pupariation and eclosion was recorded, as was the percentage of larva that developed into pupa and adult flies. Each data point represents an average time or percentage of twenty larva (*n* ≥ 8). Non-normally distributed data underwent transformation [Y = sqrt(Y) for time data and Y-arcsin (Y) for percent data]. To determine the effect of genotype and MnP administration, two-way ANOVA were performed on transformed time data, while mixed-effects analyses were performed on transformed percentage data. All data underwent Šidák’s multiple comparisons tests (GraphPad Prism 10; GraphPad Software, San Diego, CA, USA).

### 2.4. Analysis of Low-Molecular-Weight Thiols/Disulfides Using Liquid Chromatography Tandem Mass Spectrometry

On days four to six post-eclosion, about twenty heads per condition were harvested in 15% methanol supplemented with 20 mM NEM, snap frozen in liquid nitrogen, and stored at −80 °C. The samples were homogenized in a 20 mM NEM solution (15% aqueous methanol) and incubated at room temperature for 45 min. A 30 µL aliquot of each sample or calibration standard was spiked with 10 µL of mixed internal standard solution and 120 µL of acetonitrile (0.1% (*v*/*v*) formic acid) to extract the analytes. Following centrifugation (12,000× *g*, 10 min at 4 °C), 5 µL of the resulting supernatant was injected into the LC-MS system for quantitative analysis.

The LC-MS system consisted of a SCIEX ExionLC UHPLC coupled with a Sciex QTRAP 6500+ mass spectrometer (Foster City, CA, USA) equipped with an electrospray ionization source. Chromatographic separation and mass spectrometric analysis was performed as it was previously described [20,49]. Briefly, Intrada Amino Acid column (50 mm × 2 mm, 3 µm; Imtakt Corporation, Kyoto, Japan) was used along with mobile phase consisting of 25 mM ammonium formate in a 4:1 acetonitrile-water mixture (Phase A) and 0.02% formic acid in acetonitrile (Phase B). The separation was achieved using an isocratic mode followed by a gradient washing step with a total run time of 5.2 min. Mass spectrometric analysis was carried out at an ionizing voltage of 2500 V and the ion source temperature set to 550 °C. Multiple reaction monitoring was employed to monitor the following transitions for analyte detection: cysteine-NEM (*m*/*z* 247.1 → 126.0), cystine (*m*/*z* 241.2 → 152.1), GSH-NEM (*m*/*z* 433.0 → 304.0), methionine (*m*/*z* 150.0 → 56.0), CG-NEM (*m*/*z* 304.1 → 201.0), γ-glutamylcysteine (*m*/*z* 376.1 → 247.1), and cystathionine (*m*/*z* 223.0 → 134.0). Calibration curve range for GSH, cysteine, methionine, cysteinyl glycine and γ-glutamylcysteine was 0.04–400 µM, while 0.01–20 µM range was used for cystine and cystathionine. All results were normalized against protein content in homogenate. Analyst Software version 1.7 (Foster City, CA, USA) was used to perform data acquisition and analysis. Each analytical data point indicates one pooled sample of about twenty brains (*n* ≥ 6 pooled samples per condition). The effect of genotype and sex on MnP brain levels was determined using a two-way ANOVA followed by Tukey’s multiple comparison’s tests. The effect of MnP on low-molecular weight thiol levels was determined using a one-way ANOVA followed by Dunnett’s multiple comparisons tests.

### 2.5. Liquid Chromatography Tandem Mass Spectrometry Analysis of MnP Levels

MnPs were measured in homogenate using LC-MS/MS as described in [50] with some modifications. Briefly, 30 uL of homogenate was spiked with 90 µL of 0.5% acetic acid in methanol containing internal standard (MnTnHex-2-PyP^5+^). After centrifugation (12,000× *g*, 10 min at 4 °C), 5 µL of the resulting supernatant was injected into the LC-MS system for quantitative analysis. Chromatographic separation was performed on Phenomenex (Torrance, CA, USA) Kinetex™ F5 (100 mm × 2.1 mm, 2.6 µm) column, maintained at 40 °C, with the autosampler set to 5 °C. The mobile phase consisted of 0.2% heptafluorobutyric acid in water (Phase A) and methanol (Phase B). The following transitions were monitored for analyte detection: MnE (*m*/*z* 713.1 → 499.2) and MnBuOE (*m*/*z* 857.2 → 643.2). Calibration curve range was from 2 to 1000 nM for both analytes. All results were normalized against protein content in homogenate.

### 2.6. Protein Assay

Homogenate protein concentrations were measured using the Pierce Coomassie PlusProtein Assay Kit (Pierce Biotechnology, Rockford, IL, USA). Absorbance at 595 nm was recorded using the SPARK microplate reader (Tecan Group Ltd., Männedorf, Switzerland). The levels of low-molecular-weight thiols and disulfide and MnPs measured by LC-MS/MS were normalized to the protein concentration.

### 2.7. Drosophila Brain Dissection and Immunofluorescence

On days four to six post eclosion, control and parkin-null flies expressing mito-roGFP2-Orp1 or mito-roGFP2-Grx1 were anesthetized with carbon dioxide, and brains were dissected in a 2 mM NEM 1× phosphate-buffered saline solution (ThermoFisher Scientific, Waltham, MA, USA). Formaldehyde-fixed (3.7%) brains were washed with 0.3% Triton X-100 in phosphate-buffered saline (PBT) and blocked in 10% goat serum (ThermoFisher Scientific; Invitrogen, Carlsbad, CA, USA). Brains were incubated at 4 °C overnight in 1.0% anti-TH antibodies (Invitrogen, Carlsbad, CA, USA). After washing away unbound antibodies, brains were blocked again and incubated in 0.5% Alexa 594 Goat Anti-Rabbit IgG secondary antibodies for two hours (Invitrogen, Carlsbad, CA, USA). Washed brains were mounted using Invitrogen™ ProLong™ Diamond Antifade Mountant (ThermoFisher Scientific; Invitrogen, Carlsbad, CA, USA) mounting media and cured for 30 min at ambient temperature then stored at −20 °C for imaging the following day.

### 2.8. PPL1 Image Capture and Analysis

Z-stacks of one PPL1 region per fly were captured at 630× magnification with a Stellaris confocal microscope (Lecia Microsystems, Wetzlar, Germany) and analyzed with Image Pro 11 software (Media Cybernetics, Inc., Rockville, MD, USA). When roGFP2 is oxidized, its maximal excitation wavelength shifts reversibly from 488 nm to ~405 nm. Fusion of mito-roGFP2 to yeast Orp1 or human Grx1 allows measurement of relative hydrogen peroxide levels and Grx activity, respectively [44,45]. Total volumes of roGFP2 fluorescence excited by 405 nm and 488 nm lasers within the TH-labeled volume were calculated. Image Pro-generates “isosurfaces” for fluorescence emission volumes above a consistent intensity threshold to represent fluorophore expression. To determine relative levels of hydrogen peroxide and G-SH redox equilibrium (GRE), the sum of the total volume of oxidized roGFP2 per region was divided by that of the non-oxidized roGFP2. Students’ t tests or Mann–Whitney tests (for non-normally distributed data) were performed to determine the effects of genotype and MnP administration (GraphPad Prism 10; GraphPad Software, San Diego, CA, USA). Each data point represents a ratio for one PPL1 (*n* ≥ 8).

### 2.9. Climbing Assay

Climbing assays were performed by placing individual day five post-eclosion control and parkin-null flies into vertically oriented transparent polycarbonate tubes. Seventeen infrared beams pass through each tube held in a MB5 Multibeam Activity Monitor [51] (TriKinetics Inc., Waltham, MA, USA); the distance between the first and last beam is 51 mm. Yarn is placed within the first millimeters of the top and bottom of the poly-carbonate tubes to keep the fly within the detection zone. A count is recorded each time a fly crosses a beam every second for twenty minutes. A “climbing attempt” is reported each time a fly initiates an ascent. “Height climbed” is the distance of a flies’ continuous trajectory toward a higher position. A new “attempt” and “height climbed” is recorded each time a fly ascends after moving downward. The total height climbed in a twenty-minute recording period was measured and divided by number of climbs to calculate average height climbed. Each data point represents activity of one fly (*n* ≥ 27). Results were analyzed by one-way ANOVA to determine the effect of MnP administration (GraphPad Prism 10; GraphPad Software).

### 2.10. Survival Assay

Control and parkin-mutant stocks were placed on standard food supplemented with 0 or 10 µM MnE or MnBuOE. Within two days of eclosion, control (*park*^+/+^) or parkin-null (*park*^−/−^) progeny were placed in fresh treated food vials and transferred to new treated food vials once per week until all flies expired (*n* ≥ 62). Ninety-five percent prediction limits for untreated flies are plotted and the corresponding EC_50_ values were calculated to set the limits for determining statistical significance. EC_50_ values above those for the lower prediction limit or below those for the upper prediction limit are reported to be significantly different when compared to untreated flies [52].

## 3. Results

### 3.1. MnBuOE Improves Efficiency of Parkin-Null Drosophila Eclosion from Pupa Cases

Parkin-null *Drosophila* development is delayed and less efficient than in control flies (Figure 1) [6]. Thus, to screen for potential efficacy of MnPs, we placed twenty control and parkin-null third instar larva on standard food treated with 2 and 10 µM MnBuOE and observed that the number of pupa cases formed (pupariation) and the time it took for pupariation to occur was unaffected (Figure 1A). We then measured the time from pupariation to eclosion from pupa cases as well as the number of adults that eclosed and found that 10 µM MnBuOE administration increased eclosion efficiency (Figure 1B). There was no observed difference in eclosion time between control and parkin-null flies, and MnBuOE had no impact on this observation.

### 3.2. MnPs Selectively Decrease Oxidative Stress in Control Drosophila Brains

We developed an LC-MS/MS protocol to measure MnE and MnBuOE in the adult fly brain and found that, while there was no effect of genotype or sex on brain penetration, more MnBuOE accumulated in brains of flies at four-to-six days post eclosion (Figure 2A). We previously reported that GSH levels are elevated and the ratio of oxidized (cystine) to non-oxidized cysteine was decreased in brains of parkin-null flies, especially in males [53]. Levels of glutathione disulfide (oxidized glutathione) were below detection limits. These unexpected observations were accompanied by increased oxidative stress markers in PPL1 mitochondria of parkin-null flies [53]. Thus, we proposed that the whole brain compensates for the absence of parkin by increasing antioxidant responses, and the lack of this response in PPL1 contributes to its vulnerability. Here we found that MnE and MnBuOE decreased the cystine/cysteine ratio only in control fly brains, which may result from MnE/MnBuOE-driven increase in antioxidant enzyme expression via Nrf2 pathway activation as reported in hematopoietic stem progenitor cells and ovarian cancer cells (Figure 2B) [54,55]. MnPs had no effect in parkin-null flies, perhaps because the baseline cystine/cysteine ratio is so dramatically decreased (Figure 2B). We then determined that this decrease primarily resulted from MnP-mediated decreases in (oxidized) cystine (Figure 2C). Only administration of the less penetrable MnE reduced brain (non-oxidized) cysteine levels in a sexually dimorphic manner (Figure 2C). To determine whether MnP administration affects cysteine synthesis or metabolism, we measured brain levels of cysteine precursors methionine, cystathionine, and cysteinyl-glycine (cysteine-gly) as well as cysteine metabolites, gamma-glutamyl cysteine, and GSH and detected no effect of MnE or MnBuOE (Figure 3, Appendix A).

### 3.3. MnP Administration Increases Parkin-Null PPL1 Mitochondrial Antioxidant Activity in a Sexually Dimorphic Manner

We previously reported elevated hydrogen peroxide levels and GRE in males and parkin-null PPL1 mitochondria [20,53]. Here we found that the less penetrable MnE decreased hydrogen peroxide levels in parkin-null males, but did not affect parkin-null GRE (Figure 4). Interestingly, MnE had no effect on control hydrogen peroxide, but it increased GRE in female control flies and decreased it in males, in which baseline GRE is elevated (Figure 4) [53].

Following MnBuOE administration, PPL1 mitochondrial hydrogen peroxide levels were decreased in parkin-null females and control flies of both sexes, and GRE was increased in female control and parkin-null flies with no effects observed in male flies (Figure 5).

### 3.4. The More Brain- and Mitochondria-Penetrable MnP Increases Motivated Behavior in Parkin-Null Females

We measured climbing motivation (climbing attempts) and capability (height climbed) in flies raised on MnE or MnBuOE and found that the latter increased motivated behavior in parkin-null females (Figure 6A), while neither compound affected climbing capability (Figure 6B). Thus, the increased antioxidant activity in parkin-null female PPL1 mitochondria (Figure 5) can be associated with improvements in PPL1-mediated motivated behavior. Improved mitochondrial antioxidant capacity is presumably due to higher mitochondrial levels of MnBuOE. Such presumption is based on our study where MnBuOE localized in mouse brain mitochondria [41]. Interestingly, increased antioxidant activity in parkin-null male PPL1 mitochondria was not associated with similar improvements in motivated behavior, perhaps because parkin-null males have higher baseline levels of activity [53]. We also found that prolonged continuous MnBuOE, but not MnE, administration reduces median lifespan in parkin-null flies (Figure 6C). These data are congruent with higher safety/toxicity profile of MnE relative to MnBuOE [56,57]. These findings may also imply that the food concentration of the more penetrable MnBuOE achieved toxic levels within brain; therefore, future experiments will explore whether non-continuous exposure and/or lower dosing can improve parkin-null fly longevity. Interestingly, both MnPs increased the median lifespan of control *Drosophila* but did not improve climbing behavior (Supplemental Appendix A).

## 4. Discussion

We previously reported that while oxidative stress markers are elevated in parkin-null fly PPL1 mitochondria, they are decreased at the whole-brain level, especially in males [53]. Here we show that administration of a highly redox-active, brain- and mitochondria-penetrable MnBuOE [41] improves the parkin-null PPL1 mitochondrial redox environment, and that these improvements are associated with increased PPL1-driven motivated behavior in a sexually dimorphic manner. Interestingly, exposure to both MnPs reduced whole-brain oxidative stress and extended lifespan in control flies only. MnBuOE-mediated alterations in control PPL1 mitochondrial redox status were also sexually dimorphic but were not reflected in motivated behavior.

MnBuOE more readily accumulates in the mouse brain compared to MnE [42]. Here we report similar results in *Drosophila*, regardless of sex or genotype (Figure 2A). Moreover, MnBuOE, but not MnE, enters brain mitochondria at 2:1 mitochondria to cytosol ratio, while MnE is preferentially distributed in cytosol [41]. Other redox-active MnPs demonstrate brain bioavailability and attenuate the effects of 1-methyl-4-phenyl-1,2,3,6-tetrahydropyridine on mouse brain oxidized G-SH and striatal dopamine [58]. Here, increased MnBuOE brain penetrance is associated with improvements in parkin-null female motivated behavior, as well as decreased PPL1 mitochondrial hydrogen peroxide levels and increased Grx activity (Figure 5).

In contrast, MnPs failed to improve motivated behavior in parkin-null males, which generally perform better than females in climbing assays regardless of parkin mutation status [53,59], and with the exception of MnE-mediated reduction in hydrogen peroxide levels (Figure 4), they did not affect male PPL1 mitochondrial redox status. These observations may be a consequence of elevated baseline male PPL1 hydrogen peroxide and Grx activity [53]. We previously reported that folic acid administration improves motivated behavior in mixed-sex parkin-null flies, and these improvements were accompanied by decreased PPL1 mitochondrial hydrogen peroxide and Grx activity [20]. Collectively, these data indicate that behavioral improvement is particularly associated with decreases in PPL1 hydrogen peroxide, which can occur independently of changes in G-SH/Grx activity. In the present study, MnE decreased hydrogen peroxide levels in parkin-null males and MnBuOE decreased hydrogen peroxide levels in parkin-null females; however, only parkin-null females showed increased motivated behavior following MnBuOE treatment. These data suggest that female motivated behavior may be more sensitive to changes in the redox environment of PPL1 mitochondria.

The ability of both MnPs to interact with hydrogen peroxide is increasingly recognized as central to their therapeutic effects in diseases characterized by oxidative stress [40]. Multiple studies have demonstrated that MnPs function as catalytic oxido-reductants by interacting with hydrogen peroxide through both glutathione peroxidase-like and catalase-like mechanisms [35,40,60]. In the presence of cellular reductants such as G-SH and hydrogen peroxide, MnPs modulate redox-sensitive signaling pathways, such as NF-kB and Nrf2/Keap1, (via oxidation of their cysteines) and in turn protect against oxidative damage [35,40]. Further, in a catalase-like fashion MnPs also catalyze the dismutation of hydrogen peroxide into water and oxygen, further contributing to the reduction of intracellular peroxide levels and overall oxidative stress [35,40]. Mechanistically, MnPs react with hydrogen peroxide to form high-valent manganese-oxo intermediates, which are subsequently reduced by intracellular low molecular weight thiols (e.g., G-SH and cysteine), producing disulfides (e.g., glutathione disulfide and cystine). During this one-electron transition process, thiyl (such as glutathiyl) radicals are generated which can also oxidize/S-glutathionylate redox sensitive protein cysteines. As a result, in addition to direct ROS scavenging ability, MnPs also modulate cellular signaling by oxidizing/S-glutathionylating critical cysteines on transcription factors—a mechanism particularly relevant in pathologies like cancer and neurodegeneration where dysregulated ROS signaling is prominent [35,40].

Crucial molecular targets of these MnP-driven modulations are the NF-κB and Nrf2 signaling cascades, where MnP-mediated oxidation/S-glutathionylation of NF-kB and Keap1 cofactor of Nrf2 lead to decreased transcription of pro-inflammatory cytokines and increased transcription of antioxidative enzymes, respectively, resulting in attenuation of chronic inflammation [54,55,61,62]. Recent findings also suggest that MnPs facilitate the oxidation of reactive sulfur species to polysulfides in a hydrogen peroxide-dependent manner, further reducing hydrogen peroxide levels and enhancing cellular resilience to oxidative insults [40]. Collectively, these studies highlight that MnPs can lower hydrogen peroxide levels through multiple mechanisms, involving both direct detoxification of peroxides and modulation of redox-sensitive cellular pathways, which together contribute to their broad therapeutic potential in oxidative stress-related diseases.

Our data show that the less penetrable MnE only affects the control fly Grx activity (as indicated by shifts in mito-roGFP2-Grx1 emission) and it does so in a sexually dimorphic manner. The mitochondria-penetrable MnBuOE increases Grx activity in females of both genotypes. The mechanisms by which MnPs influence Grx activity are unclear. Our observations may reflect increased cellular Grx levels and antioxidant capacity resulting from protein *S*-glutathionylation and Nrf2 activation, which are key MnBuOE and Grx functions. In turn MnBuOE and Grx may potentiate the activity of each other [35,54,63]. However, the NF-kB pathway may be involved also as NF-kB gets *S*-glutathionylated by MnBuOE, under the conditions of oxidative stress, which may in turn activate the Grx through the process of de-glutathionylation [35,40,64]. The complexity of redox signaling pathways in *Drosophila* under physiological conditions (control fly) and pathological conditions (parkin-null fly), where MnBuOE, Nrf2, Grx and NF-kB may interact, will be addressed in future studies.

We observed MnP-mediated decreases in selective oxidative stress markers in young flies (four to six days post eclosion); however, we discovered that long-term consistent exposure to MnBuOE increased parkin-null fly mortality. Interestingly, both MnPs increased median lifespan of control flies. The decrease in control fly brain cystine/cysteine ratio data implies that the MnP antioxidant activity drives increases in survival of flies with homeostatic redox environments. We demonstrated that improvements in survival did not occur as a result of increased GSH synthesis. Thus, we propose that MnP-mediated decreases in the cystine/cysteine ratios are due to changes in the activity of antioxidant enzymes such as Grx- or Nrf2-mediated increases in antioxidant enzyme expression. We have previously proposed that the dimorphic redox status of PPL1 relative to the whole brain reflects a whole-brain antioxidant response that is lacking in PPL1 dopaminergic neurons in the absence of parkin function [53]. Thus, the absence of MnP-mediated changes in parkin-null cystine/cysteine ratios could occur if the brain has achieved an upper limit of antioxidant defense. MnP exposure may reduce oxidative stress in PPL1, thereby improving motivated behavior; however, long-term exposure may be selectively toxic in tissues that are intrinsically capable of boosting antioxidant capacity in the absence of parkin.

## 5. Conclusions

We have demonstrated that a relatively more brain- and mitochondrial-penetrant MnP improves only female motivated behavior in a powerful model of PD. Because long-term MnP exposure results in toxicity, future studies will address whether decreased MnP exposure and administration of newer generation MnPs could improve lifespan and motivated behavior in parkin-null flies of both sexes. MnPs directly scavenge many ROS, such as superoxide, peroxynitrite and hydrogen peroxide. They also modulate metabolic pathways by catalyzing hydrogen peroxide-driven oxidization of NF-kB and Nrf2/Keap1 transcription factor cysteines. We have provided evidence that the latter is their predominant mode of action (reviewed in [35]). Additionally, exploration of redox-active compounds with varying bioavailability and reaction favorability in parkin-null flies is warranted. Administration of compounds like Mn salens, MitoQ10, nitroxides, or combinations of these will illuminate relevant redox pathway targets; they also have the potential to reveal critical sex-specific redox signaling patterns [36].

## Figures and Tables

**Figure 1 antioxidants-14-01031-f001:**
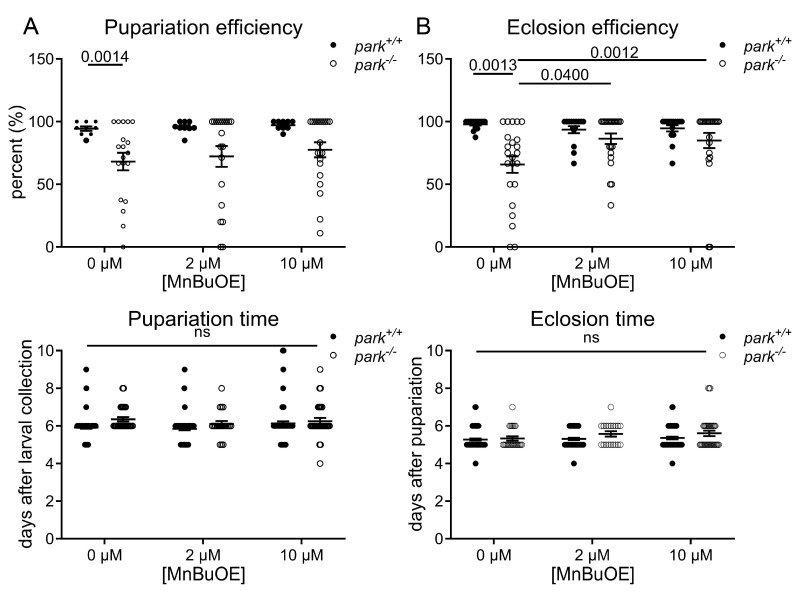
MnTnBuOE-2-PyP^5+^ improves eclosion efficiency of parkin-null pupae. Twenty control (*park*^+/+^) or parkin-null (*park*^−/−^) third instar larvae were placed on standard food supplemented with 0, 2, or 10 µM MnTnBuOE-2-PyP^5+^ (MnBuOE), and the percentage of pupae formed as well as the time required for pupa formation was recorded (**A**). The percentage of adult flies that eclosed from pupa cases and the time required for eclosion was also recorded (**B**). Data points indicate the percentage or average number of days for one vial of twenty larvae (*n* ≥ 8). Mixed-effect analyses (efficiency) or two-way ANOVA (time) followed by Sidak’s multiple comparisons tests were performed to determine the effect of genotype and MnBuOE administration. Means, standard errors of the mean (SEM), and Sidak’s tests *p* values are indicated. “ns” indicates no significance.

**Figure 2 antioxidants-14-01031-f002:**
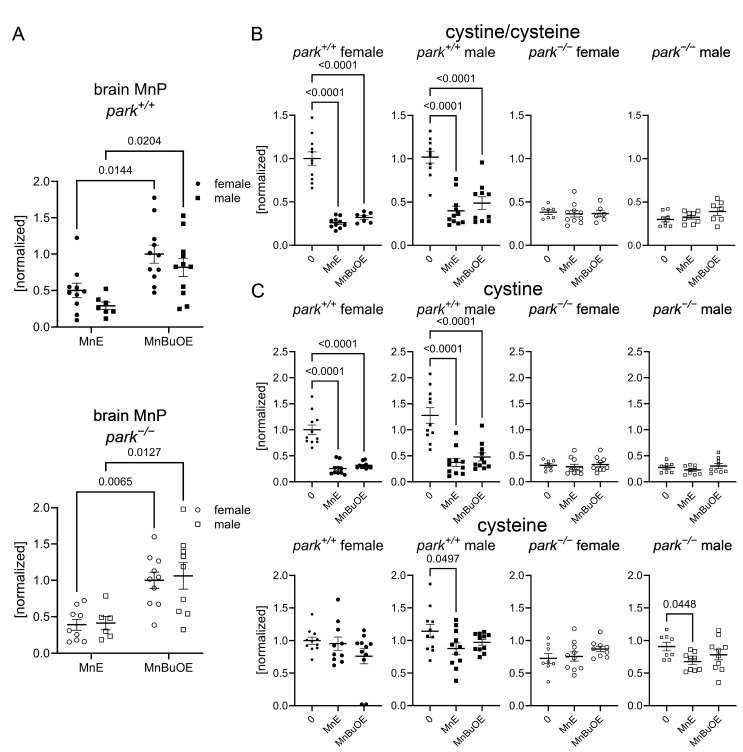
MnPs reduce an oxidative stress marker in whole brains of control *Drosophila*. Parental stocks were placed on standard food supplemented with 0 or 10 µM MnE or MnBuOE, and newly eclosed control (*park*^+/+^) or parkin-null (*park^−/−^*) progeny were collected and placed on fresh supplemented food. Brain homogenates were harvested on days four to six post-eclosion and levels of MnPs, cystine, and cysteine were measured using LC-MS/MS protocols. MnBuOE is more brain penetrant than MnE, regardless of genotype or sex (**A**). Both MnPs reduced control cystine/cysteine ratios (**B**), which are markers of oxidative stress. The reduction was driven by decreases in cystine (**C**). Male control and parkin-null brain cysteine levels were also reduced by MnE (**C**). Data points represent approximately twenty brains pooled into one brain homogenate (*n* ≥ 6). Circles represent females, and squares represent males. Filled/open circles and squares indicate control/parkin-null flies, respectively. Enlarged circles and squares indicate MnP exposure. Two-way ANOVA followed by Tukey’s multiple comparisons tests were performed to determine the effect of genotype and sex on brain MnP levels (**A**). One-way ANOVA followed by Dunnett’s multiple comparisons tests were performed to determine the effect of MnP administration on brain cystine and cysteine levels (**B**,**C**). Means, SEM, and Dunnett’s tests *p* values are indicated.

**Figure 3 antioxidants-14-01031-f003:**
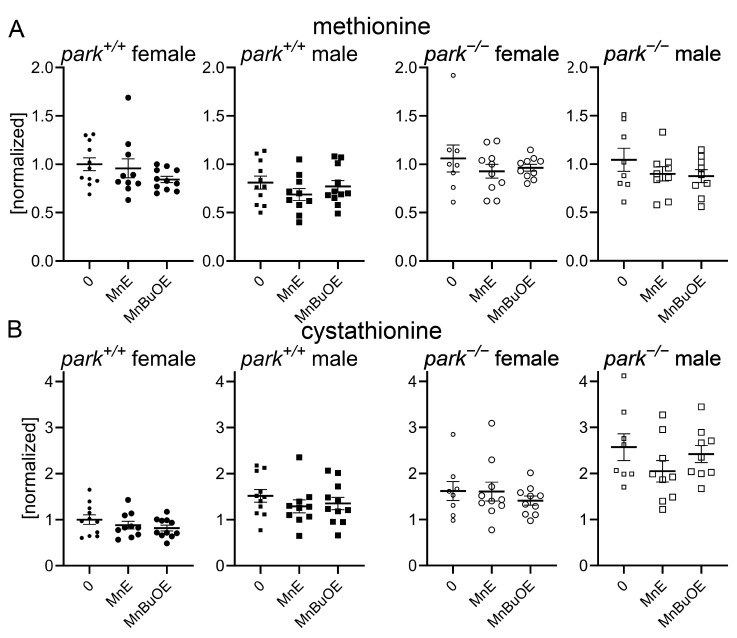
Brain levels of cysteine precursors are unaffected by MnP administration. Parental stocks were placed on standard food supplemented with 0 or 10 µM MnE or MnBuOE, and newly eclosed control (*park*^+/+^) or parkin-null (*park^−/−^*) progeny were collected and placed on fresh supplemented food. Brain homogenates were harvested on days four to six post-eclosion and methionine (**A**) and cystathionine (**B**) levels were measured using LC-MS/MS protocols. Data points represent approximately twenty brains pooled into one brain homogenate (*n* ≥ 8). Circles represent females, and squares represent males. Filled/open circles and squares indicate control/parkin-null flies, respectively. Enlarged circles and squares indicate MnP exposure. One-way ANOVA followed by Dunnett’s multiple comparisons tests were performed to determine the effect of MnP administration. Means, SEM, and Dunnett’s tests *p* values are indicated.

**Figure 4 antioxidants-14-01031-f004:**
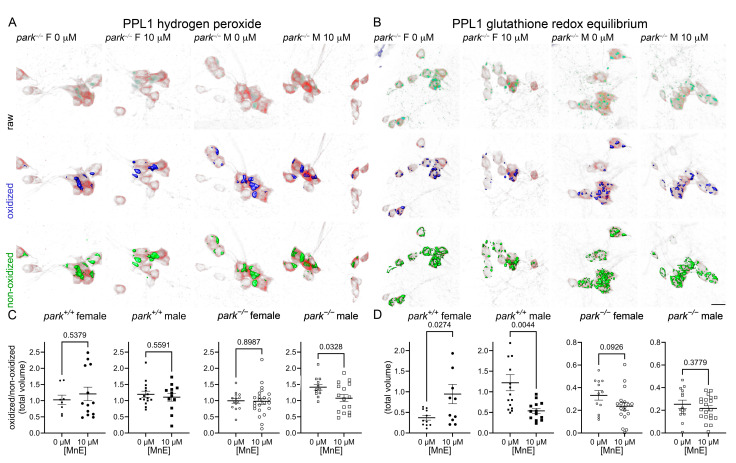
MnE reduces parkin-null male hydrogen peroxide levels and has sexually dimorphic effects on control GRE in PPL1 mitochondria. Parental stocks were placed on food supplemented with 0 or 10 µM MnE. Control (*park*^+/+^) and parkin-null (*park^−/−^*) progeny expressing TH-driven mito-roGFP2-Orp1 or mito-roGFP2-Grx1 were collected on the day of eclosion and placed on fresh supplemented food. On days four to six post-eclosion, brains were dissected, mounted, and total volumes of oxidized and non-oxidized roGFP2 emissions were calculated. Mito-roGFP1-Grx1 expression levels varied by genotype; therefore, the image capture parameters and background-removal thresholds varied by genotype. Top rows of (**A**,**B**) are representative raw images of parkin-null female and male PPL1 expressing mito-roGFP2-Orp1 (**A**) or mito-roGFP2-Grx1 (**B**). The middle and bottom rows of (**A**,**B**) indicate oxidized (blue) and non-oxidized roGFP2 (green) expression above thresholds, respectively. Scale bar represents 10 µm. Ratios of oxidized/non-oxidized mito-roGFP2 fused to Orp1 (**C**) or Grx1 (**D**) represent relative PPL1 mitochondrial hydrogen peroxide levels and GRE, respectively. Data points represent one PPL1 region (*n* ≥ 8). Circles represent females, and squares represent males. Filled/open circles and squares indicate control/parkin-null flies, respectively. Enlarged circles and squares indicate MnP exposure. Unpaired *t*-tests were performed to determine the effect of MnE administration. Means, SEM, and *p* values are indicated.

**Figure 5 antioxidants-14-01031-f005:**
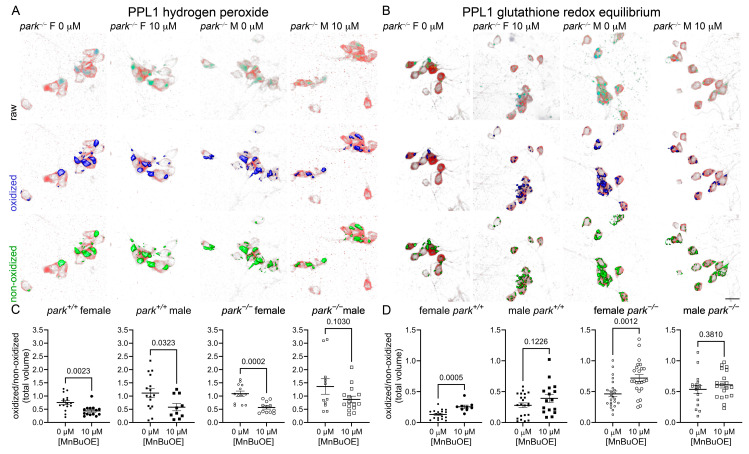
MnBuOE reduces hydrogen peroxide levels and increases female GRE in PPL1 mitochondria. Parental stocks were placed on food supplemented with 0 or 10 µM MnBuOE. Control (*park*^+/+^) and parkin-null (*park^−/−^*) progeny expressing TH-driven mito-roGFP2-Orp1 or mito-roGFP2-Grx1 were collected on the day of eclosion and placed on fresh supplemented food. On days four to six post-eclosion, brains were dissected, mounted, and total volumes of oxidized and non-oxidized roGFP2 emissions were calculated. Top rows of (**A**,**B**) are representative raw images of parkin-null female and male PPL1 expressing mito-roGFP2-Orp1 (**A**) or mito-roGFP2-Grx1 (**B**). The middle and bottom rows of (**A**,**B**) indicate oxidized (blue) and non-oxidized (green) roGFP2 expression above thresholds, respectively. Scale bar represents 10 µm. Ratios of oxidized/non-oxidized mito-roGFP2 fused to Orp1 (**C**) or Grx1 (**D**) represent relative PPL1 mitochondrial hydrogen peroxide levels and GRE, respectively. Data points represent one PPL1 region (*n* ≥ 11). Circles represent females, and squares represent males. Filled/open circles and squares indicate control/parkin-null flies, respectively. Enlarged circles and squares indicate MnP exposure. Unpaired *t*-tests were used to determine the effect of MnBuOE administration. Means, SEM, and *p* values are indicated.

**Figure 6 antioxidants-14-01031-f006:**
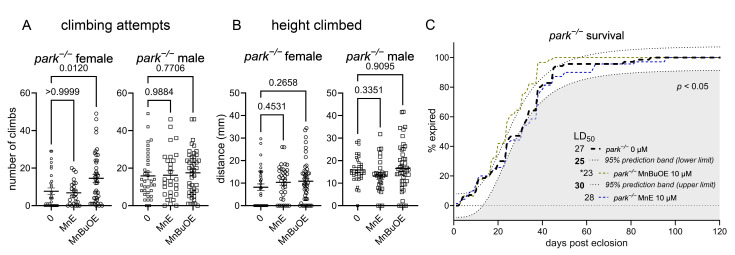
MnBuOE improves motivation to climb in female parkin-null flies. Parental stocks were placed on food supplemented with 0 or 10 µM MnE or MnBuOE, and newly eclosed control (*park*^+/+^) and parkin-null (*park^−/−^*) progeny were placed on fresh, supplemented food. Flies were transferred to the Multibeam Activity Monitor on days four to six post-eclosion, when each fly’s position in a vertical tube was recorded every second for twenty minutes. (**A**) A climbing attempt represents innate motivation to move toward a light source, and average height climbed (**B**) represents a fly’s ability to move upward in the tube. Data points represent activity of one fly (*n* ≥ 27). Circles represent females, and squares represent males. Enlarged circles and squares indicate MnP exposure. One-way ANOVA was performed to determine the effect of MnP administration (**A**,**B**). Means, SEM, and *p* values are indicated. (**C**) Continuous MnBuOE exposure reduces parkin-null fly lifespan, while MnE has no effect. The effects of MnPs are considered to be significant if the corresponding curve’s EC_50_ value is lower than that of the upper prediction limit for untreated flies (*n* ≥ 62; * indicates that *p* < 0.05) [52].

## Data Availability

The raw data supporting the conclusions of this article will be made available by the authors on request.

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
