# Peer review of "Manganese Porphyrin Reduces Oxidative Stress in Vulnerable Parkin-Null Drosophila Dopaminergic Neurons"

_antioxidants, 2025, doi:10.3390/antiox14091031_

Round 1
Reviewer 1 Report
The submission by Juba et al characterised the influence of two MnPs on parkin-null flies as a food supplement. Their objective is to compare brain and mitochondria availability, oxidative stress markers of whole brains, H2O2 and GSH of PPL1 mitochondria, and behaviour of an early onset fly PD model. Parkin is known to play a role in mitophagy, mutations in parkin impair this process causing accumulation of dysfunctional mitochondria thus oxidative stress. In view of a developmental delay in parkin-null flies, comparison on pupariation efficiency and time were first made from one of the compounds on larva and the data supports an impact on efficiency but not time. LC-MS/MS measuring MnPs, cystine/cysteine, precursors and metabolites revealed differential MnP accumulation and selective cystine reduction in control brains. Making use of a mitochondrial-bound reporter system, the authors picked up modulation of H2O2 and GSH upon MnPs administration. Finally, improved climbing ability was observed upon the treatments.
This adds to other studies in addressing a translational gap between the window of controlling oxidative stress in disease conditions in the brain, and how to safely and effectively deliver it. Including this, Lori's group has been publishing original studies on the role of parkin in neuroinflammation using a fly model.
The data is worth publishing but a key underlying question to answer is whether the compounds serve predominantly as ROS scavengers or they trigger biological response in oxidative defence.
Minor points:
1. any chance the LC-MS/MS was conducted on other tissues such as blood?
2. Include and discuss: 10.1016/j.taap.2017.04.004 and 10.3390/antiox13121444
_
Author Response
The data is worth publishing but a key underlying question to answer is whether the compounds serve predominantly as ROS scavengers or they trigger biological response in oxidative defence.
We thank the reviewer for raising the insightful concern. MnPs scavenge ROS and trigger biological oxidative defenses. They directly scavenge many ROS, such as superoxide, peroxynitrite and hydrogen peroxide. They also modulate major metabolic pathways by catalyzing hydrogen peroxide -driven oxidization of NF-kB and Nrf2/Keap1 transcription factor cysteines. We have provided evidence that the latter is their predominant mode of action (reviewed in Antioxid Redox Signal. 2018 Dec 1;29(16):1691-1724. doi: 10.1089/ars.2017.7453). We have added this information to the Conclusion section.
Minor points:
1. any chance the LC-MS/MS was conducted on other tissues such as blood?
We appreciate the reviewer’s perspective regarding effects of the MnPs on blood. Drosophilae lack a closed circulatory system; oxygen enters through trachea that are open to ambient air. They have circulating hemolymph that carries nutrients and immune cells. LC-MS/MS on Drosophila hemolymph would be very interesting; unfortunately, it is impractical compared to our pooled head protocol. Hemolymph would need to be extracted from individual flies, and only about 100 nL can be extracted per fly (Methods Protoc. 2023 Oct 12;6(5):100. doi: 10.3390/mps6050100).
2. Include and discuss: 10.1016/j.taap.2017.04.004 and 10.3390/antiox13121444.
Liang et al., 2017 (10.1016/j.taap.2017.04.004): We have added the following to lines 398 – 401. “Other redox-active MnPs demonstrate brain bioavailability and attenuate the effects of 1-methyl-4-phenyl-1,2,3,6-tetrahydropyridine (MPTP) on mouse brain GSSG and striatal dopamine (Liang et al., 2017). Here…”
Grujicic et al., 2024 (10.3390/antiox13121444): We have added the following to the Conclusion:
Additionally, exploration of redox-active compounds with varying bioavailability and reaction favorability in parkin-null flies is warranted. Administration of compounds like Mn salens, MitoQ10, nitroxides, or combinations of these will illuminate relevant redox pathway targets; they also have the potential to reveal critical sex-specific redox signaling patters (Grujicic et al., 2024).
Reviewer 2 Report
In the article entitled “Manganese porphyrin compound reduces oxidative stress in vulnerable parkin-null Drosophila dopaminergic neurons”, the authors review manganese porphyrins and the ability to reduce oxidative stress in Drosophila dopaminergic neurons.
It is a very interesting and well developed article.
Just a few points that would be important to address.
- Chronic Toxicity: Although it is recognized that MnBuOE reduces longevity in Drosophila without parkin, different exposure times or doses were not controlled to mitigate this effect.
- Little direct insight into Nrf2 or NF-κB activation.
3. Bias in drosophila sex: Although dimorphic effects are reported, the causes (e.g., differences in basal antioxidant expression or sensitivity to MnPs) are not fully explored.
N/A
Author Response
Just a few points that would be important to address.
1. Chronic Toxicity: Although it is recognized that MnBuOE reduces longevity in Drosophila without parkin, different exposure times or doses were not controlled to mitigate this effect.
We appreciate the reviewer’s pertinent comment. The objective for the current study was to use the MnP dosing strategy that was beneficial to development and climbing and determine its effect on survival. For this study, we did not include experiments exposing flies to less MnP because the less brain penetrating MnE failed to improve climbing. Nonetheless, we agree that exploring the effect of decreased MnBOE exposure on survival is warranted, and we are preparing experiments using reduced MnBOE administration as part of our next project.
2. Little direct insight into Nrf2 or NF-κB activation.
We acknowledge that we have not conducted experiments addressing effects of MnPs on Nrf2 or NF-κB activity. The results of this study have informed future work for which we are currently exploring methods to address Nrf2 or NF-κB activation in flies exposed to MnPs.
3. Bias in drosophila sex: Although dimorphic effects are reported, the causes (e.g., differences in basal antioxidant expression or sensitivity to MnPs) are not fully explored.
Our manuscript addressing the effects of sex (and loss of parkin) on whole-brain and PPL1 redox has recently been accepted for publication. In the reviewer’s version of the manuscript, we cited this as, “Juba et al., in revision.” The revised manuscript cites it as, "Juba et al., in press."